# Alcohol use among emergency medicine department patients in Tanzania: A comparative analysis of injury versus non-injury patients

**Alena Pauley**[1], **Emily C. Thatcher**[2], **Joshua T. Sarafian**[1], **Siddhesh Zadey**[1,2], **Frida Shayo**[3,4], **Blandina T. Mmbaga**[3,4,5], **Francis Sakita**[3,4], **Judith Boshe**[3,4], **João Ricardo Nickenig Vissoci**[1,2], **Catherine A. Staton**[1,2]*

**1** Duke Global Health Institute, Duke University, Durham, North Carolina, United States of America, **2** Duke Department of Surgery, Duke University Medical Center, Durham, North Carolina, United States of America, **3** Kilimanjaro Christian Medical Center, Moshi, Tanzania, **4** Kilimanjaro Christian Medical University College, Moshi, Tanzania, **5** Kilimanjaro Clinical Research Institute, Moshi, Tanzania

* catherine.lynch@duke.edu

**Data Availability Statement:** Data are only available upon reasonable request, as participants did not consent to public data publishing, and data

## Abstract

### Background

Alcohol is a leading behavioral risk factor for death and disability worldwide. Tanzania has few trained personnel and resources for treating unhealthy alcohol use. In Emergency Medicine Departments (EMDs), alcohol is a well-known risk factor for injury patients. At Kilimanjaro Christian Medical Center (KCMC) in Moshi, Tanzania, 30% of EMD injury patients (IP) test positive for alcohol upon arrival to the ED. While the IP population is prime for EMD-based interventions, there is limited data on if non-injury patients (NIP) have similar alcohol use behavior and potentially benefit from screening and intervention as well.

### Methods

This was a secondary analysis of a systematic random sampling of adult (≥18 years old), KiSwahili speaking, KCMC EMD patients surveyed between October 2021 and May 2022. When medically stable and clinically sober, participants provided informed consent. Information on demographics (sex, age, years of education, type of employment, income, marital status, tribe, and religion), injury status, self-reported alcohol use, and Alcohol Use Disorder (AUD) Identification Test (AUDIT) scores were collected. Descriptive statistics were analyzed in RStudio using frequencies and proportions.

### Results

Of the 376 patients enrolled, 59 (15.7%) presented with an injury. The IP and NIP groups did not differ in any demographics except sex, an expected difference as females were intentionally oversampled in the original study design. The mean [SD] AUDIT score (IP: 5.8 [6.6]; NIP: 3.9 [6.1]), drinks per week, and proportion of AUDIT ≥8 was higher for IP (IP:37%; NIP: 21%). However, alcohol preferences, drinking quantity, weekly expenditure on alcohol,

transfer requires a written agreement approved by Kilimanjaro Christian Medical Centre Ethics Committee and the National Institute for Medical Research (Tanzania). Data inquiries can be sent to Gwamaka W. Nselela at gwamakawilliam14@gmail.com.

**Funding:** This project was funded by the Duke Global Health Institute Graduate Student funds (AMP), and the Josiah Trent Foundation (21-06 to CAS). These two financial awards funded the salaries of JK, YS, and MMi as research assistants hired specifically for this study. The specific roles of these authors are articulated in the 'author contributions' section. No other authors received specific funding for this work. Infrastructure built by NIH grant (R01 AA027512 to CAS) was used to support the data collection process for this grant to understand gender-related aspects of alcohol use at KCMC. The funders had no role in study design, data collection and analysis, decision to publish, or preparation of the manuscript.

**Competing interests:** The authors have declared that no competing interests exist.

perceptions of unhealthy alcohol use, attempts and reasons to quit, and treatment seeking were comparable between IPs and NIPs.

## Conclusion

Our data suggests 37% of injury and 20% of non-injury patients screen positive for harmful or hazardous drinking in our setting. An EMD-based alcohol treatment and referral process could be beneficial to reduce this growing behavioral risk factor in non-injury as well as injury populations.

## Introduction

Alcohol use is a significant contributor to death and disability globally, accounting for approximately 3 million deaths and 5.1% of disability-adjusted life years (DALYs) annually [1]. This significant burden of global DALYs comes partly from the disproportionate effect on young people; in 2016, alcohol use was the primary risk factor for premature death and disability among those aged 15–49 years [1]. These numbers are underpinned by alcohol's association with injury and disease, having been linked as a causal factor in more than 200 acute and chronic diseases and injury conditions, like liver disease, cardiovascular disease, and certain cancers [2–7]. Alcohol misuse has also been associated with depression, anxiety, and phobias, with the co-occurrence of depression and alcohol misuse in particular linked to a more severe prognosis for each [8, 9]. Besides physical and mental harm, alcohol use can cause notable social and economic losses for individuals and the larger society, such as increased violence and aggressive behaviors, unsafe sexual practices, and an estimated 2.6% reduction in the global gross domestic product [10–14]. Fortunately, alcohol intake is a modifiable risk factor. As such, there is an opportunity to reduce much of the associated harm with social and behavioral changes.

In low- and middle-income countries (LMICs), alcohol-related harm has been increasing, especially within Tanzania, a country on the Eastern edge of Africa. In Tanzania, alcohol use disorder (AUD) is prevalent in 6.8% of Tanzanian citizens, which is nearly double (3.7%) the WHO African Region overall [1]. The burden associated with alcohol use is skewed by gender, with men experiencing approximately three times the attributable burden of disease compared to women [15, 16]. This proportion is seen in Tanzania as well. In 2016, the prevalence of AUDs among those aged 15 and older in Tanzania was 11.5% for men and 2.2% for women [1]. This is roughly 1.8 times the rate for men and more than 3 times the rate for women than in neighboring Malawi [1].

Moshi, a popular tourist town located at the base of Mount Kilimanjaro in Northern Tanzania, has seen a higher prevalence of alcohol use and associated disease and injury statuses in recent years [17–19]. This increase in use is in part due to a strong drinking culture and custom of early alcohol initiation in minors for members of the Chagga ethnic group, who constitute the majority of local inhabitants [20]. Standing also as contributing factors is alcohol's ready availability mixed with its low cost and recently more disposable income among inhabitants of the region [17, 20, 21]. In Moshi, high rates of alcohol use and alcohol-related harm, in combination with the lack of trained health professionals and relevant resources, lead to inadequate treatment availability [22].

As a means to reduce alcohol-related harm, alcohol-reduction interventions have been implemented in Emergency Medicine Department (EMD) settings, given the well-established association between alcohol use and injuries [23]. The consumption of alcohol impairs

cognitive functioning leading to an inhibition of perception, reflexes, and fine motor skills [24]. This inhibition can lead to more dangerous behaviors like driving under the influence, interpersonal violence, and self-harm, thereby impacting the likelihood of both intentional and unintentional injuries [23, 25]. EMDs typically stand as the first line of care for emergent injury patients. As such, EMDs tend to see a high proportion of patients with above-average regular alcohol intake [26]. At the Kilimanjaro Christian Medical Center (KCMC) in Moshi, approximately 30% of EMD injury patients (IP) tested positive for alcohol use on arrival to the EMD, a significantly larger proportion than the general population [27].

This high incidence of alcohol use has prompted the initiation of a culturally adapted Screening and Brief Intervention (BI) with Referral to Treatment (SBIRT) program among IPs at KCMC's EMD, which is currently undergoing effectiveness testing [28–30]. SBIRT is a patient-level alcohol-reduction intervention that screens patients for substance misuse, and for those identified as needing further treatment, provides a BI, which is a short, motivational interview focused on increasing awareness of patients' substance misuse and inciting positive behavioral changes [31–33]. BIs have been shown to effectively reduce alcohol intake and related consequences, especially among IPs using comparatively few resources that are scarce in LMIC settings [34–38]. IPs with AUD within KCMC's EMD were the initial patient population for the planned SBIRT. However, it is important to know if this planned intervention population is most appropriate, or if there is a need to expand intervention delivery to a larger population.

Given the already established infrastructure for the intervention, the rationale of this study is to determine the populations most at-risk for AUD within KCMC's EMD and assess if other EMD populations could benefit from an intervention. While there is data on this IP population, few, if any, studies provide data on alcohol usage among non-injury patients (NIP) in comparison to IPs. Furthermore, how and if alcohol use practices differ between IPs and NIPs is not well-studied. Given this gap in the literature and the need for reduced alcohol misuse in the area, this study aims to compare rates of alcohol use and alcohol-related behaviors between KCMC EMD IPs and NIPs to better guide alcohol-related EMD interventions.

## Methods

### Study design

This was a secondary analysis of a systematic random sampling study of KCMC EMD patients surveyed between October 2021 and May 2022. The primary study sought to describe gender differences in unhealthy alcohol use behaviors. As such, the EMD was chosen as a study location because of the high association between injuries and alcohol consumption [27, 39]. This current analysis seeks to compare alcohol consumption among IPs and NIPs who attend KCMC's EMD. Thus, demographic factors, self-reported alcohol use, and AUDIT scores were assessed between these two patient populations. The SBIRT program matrix stands as this analysis's guiding conceptual framework, given its aim of improving intervention implementation and effectiveness [40].

### Study sample

Patients were enrolled at the KCMC EMD study site, which is situated in Moshi, Tanzania. All enrolled participants met the following eligibility criteria: 1) were 18 years of age or older, 2) had the capacity to give informed consent, 3) received initial care at KCMC Emergency Department, and 4) were conversant in Kiswahili. Patients who were prisoners were excluded from our study due to increased medicolegal risk, concern for power dynamics given their

incarcerated state, and their limited access to alcohol during incarceration. For the safety of the data collection team, patients who tested positive for COVID-19 were also not approached. A patient was determined capable of providing informed consent if they were medically stabilized, clinically sober, and well enough to complete the survey verbally on their own. Those who were extremely ill or injured upon initial presentation were re-evaluated by the research team within 24 hours of arriving at KCMC or before discharge, whichever came first. Those who remained unable to consent within this time frame were excluded from study participation.

In addition to assessing gender differences, the original study aimed to compare rates of unhealthy alcohol use among females in the EMD and the reproductive health center, which required a higher proportion of female than male EMD subjects because of the lower hypothesized prevalence of alcohol misuse. Thus, the data analyzed here is representative of the EMD population when separated by gender but not when taken en masse. However, for this secondary analysis, sufficient subjects were enrolled overall to estimate the prevalence of AUDIT scores $\geq 8$ (described more in-depth below) among IPs and NIPs within a 10% margin of error and 90% confidence level. Ethical approval for this study was obtained from the Kilimanjaro Christian Medical University College Ethics Review Board, the Tanzanian National Institute of Medical Research, and the Duke University Institutional Review Board prior to the initiation of data collection.

## Data collection

Patients seeking care at KCMC's EMD were enrolled Monday through Friday at peak attendance hours, 10:00 am until 6:00 pm local time. Patients who came in overnight, were initially intoxicated, or in critical condition were followed up over 24 hours to determine if they were able to provide informed consent. If so, and if agreeing to study participation, these patients were likewise enrolled. During the data collection period, enrollment of female patients remained consistent, but enrollment of male patients was paused from January 1st, 2021, until March 31st, 2022, awaiting regulatory approval of increased male sample size. To maintain a representative, systematic random sample and meet planned enrollment goals, every female but every third male presenting to the EMD for care was approached about study participation since the EMD sees significantly more males than females.

Patients who met eligibility criteria were screened and approached once medically stabilized and offered study participation. Research assistants, who were IRB approved and trained in Good Clinical Practices, explained the research protocol and informed consent to all interested patients. Written consent was formally denoted via a signature on an informed consent form for all interested patients, except for those who were illiterate, where, depending upon their ability level, consent was provided via initials or a cross-mark. Once enrolled, patients were asked questions related to their demographics, current alcohol intake, alcohol-related behaviors, consequences of their drinking, and depressive symptoms. All data were collected in Kiswahili by a team of three Tanzanian research assistants (two female and one male). For data collection, the male research assistant surveyed all male participants, and the female research assistants surveyed all female participants. This gender-matching was done to encourage open and honest reporting of patients' experiences with alcohol based on local culture and research team experience [41]. Regulatory approval for the primary data collection project was obtained from the Kilimanjaro Christian Medical Center Ethics Committee, the Tanzanian National Institution of Medical Research (NIMR), and the Duke University Institutional Review Board. As much as possible, data was maintained in a de-identified

manner and shared by data share agreement; personal health information was used for screening and enrollment, but data were collected, stored, and analyzed in a de-identified manner.

## Variables

Information on patients' demographics (sex, age, years of education, type of employment, income, marital status, tribe, and religion), injury status, and self-reported alcohol use was collected. All demographic and self-reported alcohol use questions had previously been translated into Kiswahili by our Tanzanian research team. The Alcohol Use Disorder Identification Test (AUDIT), Drinkers' Inventory of Consequences, and the Patient Health Questionnaire 9 (PHQ-9) were the three relevant scales administered.

AUDIT is a commonly used 10-question tool for measuring alcohol consumption and alcohol-related problems, incorporating metrics that assess the quantity, frequency, and consequences of alcohol use [42, 43]. This multi-layered approach to quantifying the content of alcohol consumed makes AUDIT more sensitive and accurate in a multicultural setting, which is why it was selected as a primary measure in this study. AUDIT scores range from 0 to 40, with lower values indicating low-risk consumption and higher values suggesting alcohol dependence. Both locally and globally, scores greater than or equal to 8 are clinically significant for harmful or hazardous drinking (HHD), meaning an individual is at risk for suffering adverse health outcomes as a result of their alcohol intake [43–48]. AUDIT scores greater than or equal to 8 were thus used as a cut-off point in this analysis while also triggering the initiation of further clinical treatment for alcohol misuse.

DrInC (which ranges from 0–50) is a 50-question survey measuring alcohol-related consequences across five domains: interpersonal, intrapersonal, social consequences, impulse control, and physical [49]. While no clinically significant cut-off score exists, higher scores indicate more significant consequences for an individual [50]. Finally, PHQ-9 is a diagnostic tool used to identify the existence and severity of depression [51]. This scale ranges from 0 to 27, with higher values indicating increasingly severe depressive symptoms. Scores of 9 or greater were found to be the optimal cut-off score for identifying clinical depression in the KiSwahili-translated version of the PHQ-9 [52] and thus was the cut-off point used here. All three scales had previously been cross-culturally adapted, psychometrically validated, and clinically tested in the local context [48, 52, 53].

## Data analysis

Data on patients' demographics, injury status, self-reported alcohol use, and AUDIT, DrInC, and PHQ-9 scores were analyzed using descriptive frequencies and proportions. All variables were categorical except for measures of income, education status, and the three survey tools, which were continuous and were analyzed as means with standard deviation. AUDIT and PHQ-9 scores were dichotomized according to the cut-off values discussed previously; AUDIT scores of 8 and greater were categorized as HHD, while scores less than 8 were classified as not HHD. PHQ-9 scores of 9 and greater were categorized as a positive screen for depression, and scores less than 9 were classified as screening negative. Wilcoxon rank sum testing, Pearson's Chi-squared testing, and Fisher's exact testing were performed to determine statistical significance as appropriate. Missing data was minimal at 1 to 2 missing data points per variable, except for personal and household income as several participants were hesitant to disclose their financial status to the research staff. All data were analyzed in RStudio (version 1.4) using user-created and validated R-Packages.

## Results

### Patient demographics

During the eight months of data collection from October 1st until May 31st, 376 EMD patients were surveyed, of which 59 (15.78%) presented with injuries and 315 (84.22%) presented for non-injury-related reasons, including but not limited to fever, headache, stomachache, body numbness, or body swelling. As expected, based on our sampling strategy, females account for 70.32% of the data collected. Across all groups, most patients were Christian and from the Chagga tribe (77.54% and 46.93%, respectively). Roughly half (46.13%) of participants were employed at the time of enrollment (10.40% were students and held no other employment, and 43.20% were unemployed). This demographic data can be seen in Table 1 below.

### Alcohol use characteristics of EMD patients

In following the intention of this paper to compare rates of alcohol use and alcohol-related behaviors across IPs and NIPs presenting to KCMC's EMD, differences in AUDIT, PHQ-9, and DrInC scores were examined across these two groups. IPs had higher average [SD] AUDIT scores (5.76 [6.6]) than NIPs (3.93 [6.1]), although a sizeable proportion of individuals with HHD (AUDIT $\geq$ 8) were present in both patient groups (37.29% of IPs and 20.63% of NIPs, p<0.01). Average DrInC scores were higher in the IP population (12.92 [17.54]) than the NIP population (9.19 [15.99]), but mean PHQ-9 scores were greater among NIPs (6.60 [5.17]) than IPs (4.12 [4.14], p <0.01). Following this, a greater percentage of NIPs (21.90% of NIPs and 8.47% of IPs, p = 0.02) screened positively for depression (PHQ-9 $\geq$ 9).

Other markers of alcohol use, including drinking frequency and quantity, were also explored. AUDIT scores were higher among IPs; however, a greater proportion of NIPs appeared to drink more frequently (5.07% of NIPs but 1.7% of IPs reportedly drank every day or multiple times per day) and in unhealthy quantities (5.1% of NIPs but 3.4% of IPs drank five or more bottles per sitting). Other characteristics were comparable across both patient groups; roughly half (47.46% of IPs and 51.52% of NIPs, p = 0.93) had previously attempted to quit drinking, and a tenth (13.56% of IPs and 8.89% of NIPs, p = 0.27) had sought treatment for alcohol use, with personal reasons (39.29% of IPs and 41.72% of NIPs) cited as the primary motivation to quit. Most participants noted beer (30.51% of IPs and 23.81% of NIPs) as their alcoholic drink of choice, and 1.67% of IPs and 1.90% of NIPs spent more than 50,000 TZS (21.42 USD) on alcohol per week. To put this amount into context, the average reported personal income for the study population is 254,939.87 TZS (109.32 USD) per month or 58,674.31 TZS (25.16 USD) per week. These findings can be seen in Table 2.

This data was separated further by males and females to account for how the sex-based differences in alcohol use could impact trends among IPs and NIPs. Of those with injuries, roughly half were females (44.07%), compared to the non-injury group, where females constituted the majority (75.24%) (Table 1). Overall, males had higher average AUDIT and DrInC scores but lower PHQ-9 scores than female patients, regardless of injury status. The average AUDIT score was 7.58 [7.27] for male IPs and 6.68 [8.59] for male NIPs. This stands in comparison to females; 3.46 [5.03] and 3.03 [4.76] were the average AUDIT score for female IPs and NIPs, respectively, with p<0.01. These trends are echoed in the proportion of patients who tested positive for HHD. For males, 48.48% of those with injuries and 34.62% of those without injuries screened positively for HHD, whereas for females, 23.08% of those with

**Table 1. Demographics of EMD patients by injury status.**

| ED Demographics by Injury Status | Overall, N = 374[1] | Injury Patients N = 59[1] | Non-Injury Patients N = 315[1] | P-Value[2] |
|---|---|---|---|---|
| **Years of Education**, *missing*: 2 | 7.73 (5.05) | 8.27 (4.82) | 7.63 (5.10) | 0.16 |
| **Personal Income (TZS) per month**, *missing*: 58 | 254,939.87 (439,577.43) | 225,384.62 (316,503.07) | 259,101.08 (454,532.08) | >0.99 |
| **Household Income (TZS) per month**, *missing*: 59 | 435,968.25 (510,222.82) | 397,763.16 (492,331.32) | 441,209.39 (513,270.52) | 0.45 |
| **Female** | 263 / 374 (70.32%) | 26 / 59 (44.07%) | 237 / 315 (75.24%) | *<0.01* |
| **Religion** | | | | 0.83 |
| Christian | 290 (77.54%) | 50 (84.75%) | 240 (76.19%) | |
| Muslim | 75 (20.05%) | 8 (13.56%) | 67 (21.27%) | |
| None | 8 (2.14%) | 1 (1.69%) | 7 (2.22%) | |
| Other | 1 (0.27%) | 0 (0.00%) | 1 (0.32%) | |
| Refused/Missing | 0 / 374 | 0 / 59 | 0 / 315 | |
| **Marital Status** | | | | 0.17 |
| Divorced or Separated | 23 (6.17%) | 2 (3.39%) | 21 (6.69%) | |
| Living with a partner but not in a registered marriage | 37 (9.92%) | 9 (15.25%) | 28 (8.92%) | |
| Living with a partner in a registered marriage | 181 (48.53%) | 30 (50.85%) | 151 (48.09%) | |
| Never Married or Single | 77 (20.64%) | 13 (22.03%) | 64 (20.38%) | |
| Widowed | 54 (14.48%) | 5 (8.47%) | 49 (15.61%) | |
| Refused/Missing | 2 / 373 | 0 / 59 | 2 / 315 | |
| **Employment Status** | | | | 0.42 |
| Employed | 173 (46.13%) | 31 (52.54%) | 142 (45.08%) | |
| Unemployed | 162 (43.20%) | 22 (37.29%) | 140 (44.44%) | |
| Student | 39 (10.40%) | 6 (10.17%) | 33 (10.48%) | |
| Refused/Missing | 0 / 374 | 0 / 59 | 0 / 315 | |
| **Tribe** | | | | 0.67 |
| Chagga | 176 (46.93%) | 31 (52.54%) | 145 (46.03%) | |
| Iraq | 13 (3.47%) | 3 (5.08%) | 10 (3.17%) | |
| Maasai | 18 (4.80%) | 2 (3.39%) | 16 (5.08%) | |
| Mmeru | 18 (4.80%) | 4 (6.78%) | 14 (4.42%) | |
| Muha or Non-African | 3 (0.80%) | 1 (1.69%) | 2 (0.63%) | |
| Nyaturu | 5 (1.33%) | 0 (0.00%) | 5 (1.59%) | |
| Other African | 64 (17.07%) | 7 (11.86%) | 57 (18.10%) | |
| Pare | 44 (11.73%) | 8 (13.56%) | 36 (11.43%) | |
| Sambaa | 15 (4.00%) | 2 (3.39%) | 13 (4.13%) | |
| Sukuma | 18 (4.80%) | 1 (1.69%) | 17 (5.38%) | |
| Refused/Missing | 0 / 374 | 0 / 59 | 0 / 315 | |

[1]Mean (SD); n / N (%)

[2]Wilcoxon rank sum test; Pearson's Chi-squared test; Fisher's exact test

P-Value is a comparison between Injury and Non-injury Patients

Significant at P < 0.05

Italics indicate p < 0.05

injuries and 16.03% of those without injuries had AUDIT scores greater than or equal to 8, with p<0.01.

Males also had average DrInC scores (16.30 [19.32] for male IPs and 14.28 [20.46] for male NIPs), which were higher than their female counterparts. For females, average DrInC scores were 8.62 [14.19] for female IPs and 7.49 [13.90] for female NIPs. While males had the highest AUDIT and DrInC scores, females were more likely to screen positively for depression (those

**Table 2. Alcohol use characteristics of EMD patients by injury status.**

| EMD Alcohol Use Characteristics by Injury Status | Overall, N = 374[1] | Injury Patients N = 59[1] | Non-Injury Patients N = 315[1] | P-Value[2] |
|---|---|---|---|---|
| **AUDIT Score** | 4.22 (6.25) | 5.76 (6.66) | 3.94 (6.13) | *0.02* |
| **DrInC Score** | 9.76 (16.30) | 12.92 (17.54) | 9.17 (16.01) | *0.01* |
| **PHQ-9 Score** | 6.21 (5.11) | 4.12 (4.14) | 6.60 (5.18) | *<0.01* |
| **Alcohol Preferences** | | | | 0.91 |
| Beer | 93 (24.87%) | 18 (30.51%) | 75 (23.81%) | |
| Changaa, Dadii, Gongo, Piwa | 8 (2.14%) | 0 (0.00%) | 8 (2.54%) | |
| Light Beer | 39 (10.43%) | 3 (5.08%) | 36 (11.43%) | |
| Liquor/Spirits | 11 (2.94%) | 2 (3.39%) | 9 (2.86%) | |
| Mbege | 55 (14.71%) | 10 (16.95%) | 45 (14.26%) | |
| None | 111 (29.68%) | 15 (25.42%) | 96 (30.48%) | |
| Other | 2 (0.53%) | 0 (0.00%) | 2 (0.63%) | |
| Ulanzi | 4 (1.07%) | 1 (1.69%) | 3 (0.95%) | |
| Wine | 46 (12.30%) | 10 (16.95%) | 36 (11.43%) | |
| Refused/Missing | 5 / 374 | 0 / 59 | 5 / 315 | |
| **Drinking Frequency** | | | | *0.01* |
| 0 times/week | 119 (31.82%) | 16 (27.12%) | 103 (32.70%) | |
| 1–2 times/week | 185 (49/47%) | 24 (40.68%) | 161 (51.11%) | |
| 3–4 times/week | 47 (12.53%) | 16 (27.12%) | 31 (9.84%) | |
| 5–6 times/week | 4 (1.07%) | 1 (1.69%) | 3 (0.95%) | |
| Every day | 14 (3.74%) | 0 (0.00%) | 14 (4.44%) | |
| Multiple times a day | 3 (0.80%) | 1 (1.69%) | 2 (0.63%) | |
| Refused/Missing | 2 / 375 | 1 / 59 | 1 / 315 | |
| **Drinking Quantity** | | | | 0.27 |
| 0 drinks | 119 (31,82%) | 16 (27.12%) | 103 (32.70%) | |
| 1–2 bottles | 172 (45.99%) | 24 (40.68%) | 148 (46.98%) | |
| 3–4 bottles | 62 (16.58%) | 16 (27.12%) | 46 (14.60%) | |
| 5–6 bottles | 13 (3.48%) | 1 (1.69%) | 12 (3.81%) | |
| >6 bottles | 5 (1.34%) | 1 (1.69%) | 4 (1.27%) | |
| Refused/Missing | 3 / 374 | 1 / 59 | 2 / 315 | |
| **Weekly Alcohol Expenses (TZS)** | | | | 0.15 |
| 0–10000 | 284 (75.94%) | 38 (64.41%) | 246 (78.10%) | |
| 10001–50000 | 79 (21.12%) | 19 (32.20%) | 60 (18.99%) | |
| 50001–100000 | 7 (1.87%) | 1 (1.69%) | 6 (1.90%) | |
| Refused/Missing | 4 / 374 | 1 / 59 | 3 / 315 | |
| **Attempted Quitting Alcohol** | | | | 0.93 |
| No | 170 (45.45%) | 29 (49.15%) | 141 (44.76%) | |
| Yes | 191 (51.07%) | 28 (47.46%) | 163 (51.75%) | |
| Refused/Missing | 13 / 374 | 2 / 59 | 11 / 315 | |
| **Reason for Quitting** | | | | 0.62 |
| Family | 13 (6.81%) | 3 (10.71%) | 10 (6.13%) | |
| Financial | 19 (9.95%) | 3 (10.71%) | 16 (9.82%) | |
| Health | 66 (34.55%) | 8 (28.57%) | 58 (35.58%) | |
| Other | 1 (0.52%) | 0 (0.00%) | 1 (0.61%) | |
| Personal | 79 (41.36%) | 11 (39.29%) | 68 (41.72%) | |
| Spiritual | 13 (6.81%) | 3 (10.71%) | 10 (6.13%) | |
| Not Applicable | 183 | 31 / 59 | 152 / 315 | |
| **Alcohol Use perceived as Unhealthy** | | | | 0.93 |

*(Continued)*

**Table 2.** (Continued)

| EMD Alcohol Use Characteristics by Injury Status | Overall, N = 374[1] | Injury Patients N = 59[1] | Non-Injury Patients N = 315[1] | P-Value[2] |
|---|---|---|---|---|
| No | 86 (22.99%) | 16 (27.12%) | 70 (22.22%) | |
| Yes | 283 (75.67%) | 43 (72.88%) | 240 (76.19%) | |
| Refused/Missing | 5 / 374 | 0 / 59 | 5 / 315 | |
| **Sought Treatment for Alcohol Use** | | | | 0.27 |
| No | 335 (89.58%) | 51 (86.44%) | 284 (90.16%) | |
| Yes | 36 (9.62%) | 8 (13.56%) | 28 (8.89%) | |
| Refused/Missing | 3 / 374 | 0 / 59 | 3 / 315 | |
| **Sought Psychiatric Treatment,** *missing*: 2 | | | | *0.02* |
| No | 340 (90.91%) | 49 (83.05%) | 293 (93.02%) | |
| Yes | 31 (8.29%) | 10 (16.95%) | 21 (6.67%) | |
| Refused/Missing | 3 / 374 | 0 / 59 | 3 / 315 | |
| **AUD Status (AUDIT ≥ 8)** | 87 / 374 (23.26%) | 22 / 59 (37.29%) | 65 / 315 (20.63%) | *<0.01* |
| **Depression Status (PHQ-9 ≥ 9)** | 74 / 374 (19.79%) | 5 / 59 (8.47%) | 69 / 315 (21.90%) | *0.02* |

[1]Mean (SD); n / N (%)

[2]Wilcoxon rank sum test; Pearson's Chi-squared test; Fisher's exact test

P-Value is a comparison between Injury and Non-injury Patients

Significant at P < 0.05

Italics indicate p < 0.05

with a PHQ-9 ≥ 9 constituted 15.38% of female IPs and 25.74% of female NIPs) compared to males (3.03% of male IPs and 10.26% of male NIPs had a PHQ-9 ≥ 9). Of all groups, female NIPs had the highest average PHQ-9 score (7.30 [5.16]), followed by female IPs with an average of 6.62 [3.94]. These differences were all statistically significant (p<0.01). These reports and others can be seen in Table 3.

## Discussion

Alcohol use is a leading and growing behavioral risk factor for poor health, injury, and illness. Especially in low-resource settings, each patient who interfaces with the healthcare system represents a chance to screen and initiate primary and secondary preventative health measures [22]. This is the first analysis to compare alcohol use and alcohol-related behaviors between injury and non-injury EMD patients in Moshi, Tanzania, with the goal of describing opportunities for initiation of screening and intervention opportunities. Our evidence demonstrates that both injury and non-injury patients in Moshi have high rates of harmful and hazardous alcohol use, and non-injured patients also have high rates of psychological comorbidities. These findings indicate that future screening and intervention initiatives should focus on the entire EMD population rather than specific injury groups.

The data presented has shown that a large proportion of both IPs and NIPs presenting to KCMC's EMD screened positive for HHD and exhibited unhealthy alcohol use behaviors. While IPs (37%) had a higher prevalence of HHD than NIPs (21%), a higher proportion of NIPs drank daily or drank five or more drinks per sitting than IPs. The higher rate of AUDIT ≥ 8 among IPs likely stems from this group having a higher proportion of males (given males generally consume more alcohol [16] and the close association of alcohol use and injuries that prompt EMD visits [26]. For example, previous work by our group has found that 30% of IPs at KCMC's EMD were ETOH positive upon admission [27]. While the literature

**Table 3. Alcohol use characteristics of EMD patients by injury status and gender.**

| EMD Alcohol Use Characteristics by Injury Status and Gender | Female Injury Patients, N = 26[1] | Female Non-Injury Patients, N = 237[1] | Male Injury Patients, N = 33[1] | Male Non-Injury Patients, N = 78[1] | P-Value[2] |
|---|---|---|---|---|---|
| **AUDIT Score** | 3.46 (5.03) | 3.03 (4.76) | 7.58 (7.27) | 6.68 (8.59) | <0.01 |
| **DrInC Score** | 8.62 (14.19) | 7.49 (13.90) | 16.30 (19.32) | 14.28 (20.46) | <0.01 |
| **PHQ-9 Score** | 6.62 (3.94) | 7.30 (5.16) | 2.15 (3.15) | 4.49 (4.68) | <0.01 |
| **Alcohol Preferences** | | | | | <0.01 |
| Beer | 5 (19.23%) | 59 (24.89%) | 13 (39.39%) | 16 (20.51%) | |
| Changaa, Dadii, Gongo, Piwa | 0 (0.00%) | 5 (2.11%) | 0 (0.00%) | 3 (3.85%) | |
| Light Beer | 0 (0.00%) | 20 (8.43%) | 3 (9.09%) | 16 (20.51%) | |
| Liquor/Spirits | 0 (0.00%) | 3 (1.27%) | 2 (6.06%) | 6 (7.69%) | |
| Mbege | 5 (19.23%) | 35 (14.77%) | 5 (15.15%) | 10 (12.82%) | |
| None | 12 (46.15%) | 86 (36.29%) | 3 (9.09%) | 10 (12.82%) | |
| Other | 0 (0.00%) | 2 (0.84%) | 0 (0.00%) | 0 (0.00%) | |
| Ulanzi | 0 (0.00%) | 2 (0.84%) | 1 (3.03%) | 1 (1.28%) | |
| Wine | 4 (15.38%) | 25 (10.55%) | 6 (18.18%) | 11 (14.10%) | |
| Refused/Missing | 0 / 26 | 0 / 237 | 0 / 33 | 5 / 78 | |
| **Drinking Frequency[3]** | | | | | |
| 0 times/week | 12 (46.15%) | 87 (36.71%) | 4 (12.12%) | 16 (20.51%) | |
| 1–2 times/week | 6 (23.08%) | 120 (50.63%) | 18 (54.54%) | 41 (52.56%) | |
| 3–4 times/week | 8 (30.77%) | 18 (7.59%) | 8 (24.24%) | 13 (16.67%) | |
| 5–6 times/week | 0 (0.00%) | 2 (0.84%) | 1 (3.03%) | 1 (1.28%) | |
| Every day | 0 (0.00%) | 10 (4.22%) | 0.00 (0%) | 4 (5.13%) | |
| Multiple times a day | 0 (0.00%) | 0 (0.00%) | 1 (3.03%) | 2 (2.56%) | |
| Refused/Missing | 0 / 26 | 0 / 237 | 1 / 33 | 2 / 78 | |
| **Drinking Quantity** | | | | | <0.01 |
| 0 drinks | 12 (46.15%) | 88 (37.13%) | 4 (12.12%) | 15 (19.23%) | |
| 1–2 bottles | 8 (30.77%) | 106 (44.73%) | 16 (48.48%) | 42 (53.85%) | |
| 3–4 bottles | 6 (23.08%) | 35 (15%) | 10 (30.30%) | 11 (14.10%) | |
| 5–6 bottles | 0 (0.00%) | 7 (2.95%) | 1 (3.03%) | 5 (6.41%) | |
| >6 bottles | 0 (0.00%) | 1 (0.42%) | 1 (3.03%) | 3 (3.85%) | |
| RefusedMissing | 0 / 26 | 0 / 237 | 1 / 33 | 2 / 78 | |
| **Weekly Alcohol Expenses (TZS)** | | | | | <0.01 |
| 0–10000 | 21 (80.77%) | 200 (84.39%) | 17 (51.51%) | 46 (58.97%) | |
| 10001–50000 | 5 (19.23%) | 35 (14.77%) | 14 (42.42%) | 25 (32.05%) | |
| 50001–100000 | 0 (0.00%) | 2 (0.84%) | 1 (3.03%) | 4 (5.13%) | |
| Refused/Missing | 0 / 26 | 0 / 237 | 1 / 33 | 3 / 78 | |
| **Attempted Quiting Alcohol** | | | | | <0.01 |
| No | 15 (57.69%) | 113 (47.68%) | 14 (42.42%) | 28 (35.90%) | |
| Yes | 11 (42.31%) | 122 (51.48%) | 17 (51.51%) | 41 (52.56%) | |
| Refused/Missing | 0 / 26 | 2 / 237 | 2 / 33 | 9 / 78 | |
| **Reason for Quitting** | | | | | 0.07 |
| Family | 0 (0.00%) | 8 (6.56%) | 3 (17.65%) | 2 / 41 (4.88%) | |
| Financial | 1 (9.09%) | 6 (4.92%) | 2 (11.76%) | 10 / 41 (24.39%) | |
| Health | 5 (45.45%) | 45 (36.89%) | 3 (17.65%) | 13 / 41 (31.71%) | |
| Other | 0 (0.00%) | 1 (0.82%) | 0 (0.00%) | 0 / 41 (0.00%) | |
| Personal | 4 (36.36%) | 56 (45.90%) | 7 (41.18%) | 12 / 41 (29.27%) | |
| Spiritual | 1 (9.09%) | 6 (4.92%) | 2 (11.76%) | 4 / 41 (9.76%) | |
| Not Applicable | 15 / 26 | 115 / 237 | 16 / 33 | 37 / 78 | |

*(Continued)*

**Table 3.** (Continued)

| EMD Alcohol Use Characteristics by Injury Status and Gender | Female Injury Patients, N = 26[1] | Female Non-Injury Patients, N = 237[1] | Male Injury Patients, N = 33[1] | Male Non-Injury Patients, N = 78[1] | P-Value[2] |
|---|---|---|---|---|---|
| **Alcohol Use Perceived as Unhealthy** | | | | | *0.01* |
| No | 9 (34.62%) | 64 (27.00%) | 7 (21.21%) | 6 (7.69%) | |
| Yes | 17 (65.38%) | 168 (70.88%) | 26 (78.78%) | 72 (92.31%) | |
| Refused/Missing | 0 / 26 | 5 / 237 | 0 / 33 | 0 / 78 | |
| **Sought Treatment for Alcohol Use** | | | | | *0.04* |
| No | 24 (92.31%) | 219 (92.41%) | 27 (81.82%) | 65 (83.33%) | |
| Yes | 2 (7.69%) | 15 (6.33%) | 6 (18.18%) | 13 (16.67%) | |
| Refused/Missing | 0 / 26 | 2 / 237 | 0 / 33 | 0 / 78 | |
| **Sought Psychiatric Treatment** | | | | | 0.09 |
| No | 20 (76.92%) | 217 (91.56%) | 29 (87.87%) | 74 (94.87%) | |
| Yes | 6 (23.08%) | 18 (7.59%) | 4 (12.12%) | 3 (3.85%) | |
| Refused/Missing | 0 / 26 | 1 / 237 | 0 / 33 | 0 / 78 | |
| **AUD Status (AUDIT $\geq$ 8)** | 6 / 26 (23.08%) | 38 / 237 (16.03%) | 16 / 33 (48.48%) | 27 / 78 (34.62%) | *<0.01* |
| **Depression Status (PHQ-9 $\geq$ 9)** | 4 / 26 (15.38%) | 61 / 237 (25.74%) | 1 / 33 (3.03%) | 8 / 78 (10.26%) | *<0.01* |

[1]Mean (SD); n / N (%)

[2]Wilcoxon rank sum test; Pearson's Chi-squared test; Fisher's exact test

[3]Sample size was insufficient to find P-Value for Drinking Frequency

P-Value is a comparison between Injury and Non-injury Patients

Significant at $P < 0.05$

Italics indicate $p < 0.05$

on alcohol use among non-injury EMD patients is limited globally and in Tanzania, one study in Belgium found that patients presenting with psychiatric problems had the highest incidence of being intoxicated compared to other complaints [54]. However, as EMDs in Tanzania and Belgium differ vastly, this finding may not be generalizable to our study population.

The rates shown in our analysis are comparable to or higher than other population-wide alcohol use estimates in Moshi. In 2008, for example, Mitsunga and Larsen found that the lowest rates of alcohol abuse in Moshi were in females with partners (7.0%) and the highest rates in males (22.8%) (CAGE > 2 as the definition of alcohol abuse) [17]. Further, AUDIT screening at an outpatient primary health center in Moshi conducted by Mushi et al. showed that 23.9% of primary care patients reported AUDIT scores $\geq$ 8 [22]. When stratifying and comparing our data to Mushi's estimates by sex, a higher proportion of EMD IPs and NIPs have HHD than primary care patients. Mushi et al. found that 38.7% of males and 13.1% of females had AUDIT scores $\geq$8 [22]. In our data set, 48% of male IPs and 35% of male NIPs had AUDIT $\geq$8, and 23% of female IPs and 16% of female NIPs had AUDIT $\geq$8. These direct comparisons suggest that within Moshi, KCMC EMD patients, both IPs and NIPs, have high proportions of HHD/ AUDIT$\geq$8 compared to other local populations, demonstrating an opportunity for screening and intervention in a high-risk population.

In comparing regional data, Kenyatta National Hospital in Nairobi, using a more sensitive AUDIT-C, found 91.5% of male and 8.5% of female EMD IPs screened positive for HHD [55]. Concerningly, our higher prevalence of HHD amongst female IPs within KCMC suggests more severe alcohol misuse among this population. Similarly, in looking at data from Uganda, where 5.8% of male primary care patients screened positively for HHD via AUDIT [56], we saw a nearly 6-fold higher male AUDIT screen in our study location. This further implies a

very high concentration of patients with HHD in KCMC EMD. Across the continent in Ghana, 27% of injured patients in the EMD reported a history of harmful alcohol use as measured by AUDIT ≥ 8 [57]. In both these studies, no comparison was made to the rates in non-injured patients.

As in other settings [58–60], our patients have both AUD and other untreated mental health comorbidities. Overall, 1 in 5 or 20% of patients (8.5% of IPs and 22% of NIPs) sampled scored ≥ 9 on the PHQ9 (screening positive for depression), of which 91% had neither sought nor received treatment. Compared to our data, the WHO estimates the prevalence of depression to be approximately 4.1% within the general population of Tanzania [61] and 5.0% among the global adult population [62]. These depression screenings are five times higher in our EMD population and four times higher than the global population, highlighting the high mental health burden this patient group experiences and calling for expanded clinical services to support this high-risk group.

Accessing care for mental health disorders is limited by health literacy, stigma, and knowledge of and availability of services in resource-limited settings globally and within Tanzania [63–66]. It is unfortunately not surprising that over 90% of our population reports not seeking (nor obtaining) treatment services for either mental health or alcohol treatment services. The Tanzanian government's entire expenditure on mental health comprises 4.0% of the total government health budget, and there are 1.31 total mental health professionals per 100,000 people in Tanzania compared to 15.32 mental health professionals per 100,000 in neighboring Kenya [67]. Aside from a lack of resources, stigma and negative perceptions about mental health can limit patients seeking care. Within this already resource-strained setting, a KCMC-based study found that 71% of providers devalued or discriminated against their patients who suffered from alcohol use disorders [68]. Another study performed in Domoda, Tanzania, showed that 58.9% of the surveyed population had negative attitudes toward people with mental health problems [69]. These stigmatizing perceptions and generalizations about Tanzanians with mental health problems and substance use (including alcohol) will continue to limit cultural acceptance of seeking care.

Given the high rates of unhealthy alcohol use and mental health disorders found among our KCMC EMD IPs and NIPs, there is an opportunity to improve screening and treatment initiation for a more comprehensive screening system. The implementation of high-risk alcohol use screening like SBIRT within LMICs like Tanzania has been recommended as a way to reduce alcohol-related harm [70]. Kenya and South Africa are the only African countries to successfully use alcohol-related SBIRT programs in their EMDs [71]. However, there was an exclusive focus on IPs instead of the total EMD patient population in these instances. KCMC is another institution that has implemented similar services among EMD IPs in recent research-based intervention initiatives [28, 29].

However, because alcohol use and psychiatric disorders are present at high rates in our study population and are commonly co-occurring [8], initiating both psychological and substance misuse-based screenings among all-comers in this setting represents an opportunity to better assist disordered patients as a whole. Similar models could be translated to this setting with a limited increase in resource utilization as basic screening measures are already being utilized among EMD IPs. As seen in this KCMC EMD population, a high prevalence of alcohol use and mental health disorders was found regardless of their injury status. Implementing screening measures for both excessive alcohol consumption and depression at this clinical site could both increase the identification of patients with high-risk alcohol and mental health needs and subsequently facilitate the delivery of appropriate treatments for high-risk patients. Expanding the scope of these current measures to incorporate NIPs at KCMC and screening

for depression could lead to improved health outcomes for more patients without having to expend significantly more resources.

In summary, both IPs and NIPs presenting to KCMC's EMD in Moshi, Tanzania, have higher rates of alcohol use disorder and depression than the general population. Given these disorders' heavy detrimental effects on patient health outcomes, more robust screening and treatment services are needed at KCMC's EMD. Local policy efforts should be aware of the significant mental health burdens found among this patient population and the potential benefit of implementing expanded screening services.

## Strengths and limitations

Several important factors should be considered when interpreting the data presented in this paper. The purposeful oversampling of female patients in the study from which this data originates impacted our reported incidence of alcohol use when looking at amalgamated IP and NIP statistics. This resulted in an underestimation of true IP and NIP alcohol intake in Table 2. To account for this bias, we have also reported statistics separated by gender, as seen in Table 3. Further, a complete registry of all patients who present for services at KCMC is unavailable, which restricts our ability to evaluate how representative this sample is of the EMD's total population. While this limitation is offset by our large sample size and systematic random sampling strategy, it should still be taken into account. Our data set is also limited in terms of specificity around patients' chief complaints. This constrains our ability to compare complaint-specific groups, which could help assess the risk of alcohol misuse in particular EMD patient groups. Therefore, we recommend that future studies look more closely at EMD patients' demographics and chief complaints in combination with their alcohol intake, as it might vary with each unique setting and characteristic.

Some additional factors may have altered the composition of enrolled patients from what may ordinarily be seen of the full EMD population. First, our data was collected throughout several waves of the COVID-19 pandemic. The limited resources available to protect KCMC's EMD patients from the spread of COVID-19 may have dissuaded those with lower acuity complaints from seeking care at the EMD, leading our sample to encompass a higher proportion of patients with severe conditions. Further, patients who remained unable to consent over 24 hours were excluded from study participation. While this was deemed necessary for ethical and safety reasons, it may have impacted study demographics given potentially different alcohol use patterns in this patient group than those who could provide consent. Last, the data collected from participants was dependent upon the accuracy of their self-reported demographic and alcohol intake. While several measures were taken throughout the study to encourage truthful reporting of patients' typical alcohol intake (including gender-matching data collectors with participants and collecting all data in private, confidential environments), there may have been an underreporting of true alcohol consumption. This may be due to the societal stigmatization surrounding heavy alcohol use in Tanzania, which encourages secretive alcohol use behaviors, especially among females.

More research is needed to identify better treatments that would be effective for all EMD patients presenting with unhealthy alcohol use. Building upon the information presented here, to reduce the burden caused by alcohol in this region, more effective, evidence-based care options are urgently needed.

## Conclusion

Of all patients who presented to KCMC's EMD, we found that 37% of IPs and 21% of NIPs scored ≥ 8 on the AUDIT. As this score is clinically significant for hazardous or harmful

alcohol use, our findings suggest that a significant proportion of both patient populations could benefit from an EMD-based alcohol treatment and referral process to reduce this growing behavioral risk factor. This is especially important given this region's lack of alcohol-related treatment options and trained personnel. We intend for this data to be used to shape and inform research and treatment programs related to alcohol and reduce the burden of alcohol use in Moshi, Tanzania.

## Supporting information

**S1 File. Duke IRB approval.**
(PDF)

**S2 File. KCMC IRB approval.**
(PDF)

**S3 File. NIMR IRB approval.**
(PDF)

**S1 Checklist. STROBE checklist.**
(DOCX)

**S1 Questionnaire. Inclusivity in global research.**
(DOCX)

## Author Contributions

**Conceptualization:** Frida Shayo, Blandina T. Mmbaga, Francis Sakita, Judith Boshe, Catherine A. Staton.

**Data curation:** Alena Pauley.

**Formal analysis:** Frida Shayo, Francis Sakita, Catherine A. Staton.

**Funding acquisition:** Alena Pauley, Catherine A. Staton.

**Methodology:** Alena Pauley, João Ricardo Nickenig Vissoci, Catherine A. Staton.

**Supervision:** Blandina T. Mmbaga, João Ricardo Nickenig Vissoci, Catherine A. Staton.

**Writing – original draft:** Alena Pauley, Emily C. Thatcher, Joshua T. Sarafian, Siddhesh Zadey.

**Writing – review & editing:** Alena Pauley, Emily C. Thatcher, Joshua T. Sarafian, Siddhesh Zadey, João Ricardo Nickenig Vissoci, Catherine A. Staton.

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
