## [Decision Letter · Decision Letter 0]

19 Jun 2023

PGPH-D-23-00562

Alcohol Use among Emergency Medicine Department Patients in Tanzania: A Comparative Analysis of Injury Versus Non-injury Patients

Dear Dr. Staton,

Thank you for submitting your manuscript to PLOS Global Public Health. After careful consideration, we feel that it has merit but does not fully meet PLOS Global Public Health’s publication criteria as it currently stands. Therefore, we invite you to submit a revised version of the manuscript that addresses the points raised during the review process.

We look forward to receiving your revised manuscript.

Kind regards,

Hani Mowafi, M.D., M.P.H.

Academic Editor

Journal Requirements:

1. Please include a complete copy of PLOS’ questionnaire on inclusivity in global research in your revised manuscript. Our policy for research in this area aims to improve transparency in the reporting of research performed outside of researchers’ own country or community. The policy applies to researchers who have travelled to a different country to conduct research, research with Indigenous populations or their lands, and research on cultural artefacts. The questionnaire can also be requested at the journal’s discretion for any other submissions, even if these conditions are not met.  Please find more information on the policy and a link to download a blank copy of the questionnaire here: https://journals.plos.org/globalpublichealth/s/best-practices-in-research-reporting. Please upload a completed version of your questionnaire as Supporting Information when you resubmit your manuscript.”

2. Please provide additional details regarding participant consent. In the ethics statement in the Methods and online submission information, please ensure that you have specified what type you obtained (for instance, written or verbal, and if verbal, how it was documented and witnessed).

3. Please amend your detailed Financial Disclosure statement. This is published with the article. It must therefore be completed in full sentences and contain the exact wording you wish to be published.

Additional Editor Comments (if provided):

Reviewers' comments:

Reviewer's Responses to Questions

**Comments to the Author**

1. Does this manuscript meet PLOS Global Public Health’s publication criteria? Is the manuscript technically sound, and do the data support the conclusions? The manuscript must describe methodologically and ethically rigorous research with conclusions that are appropriately drawn based on the data presented.

Reviewer #1: Yes

Reviewer #2: Yes

Reviewer #3: Yes

2. Has the statistical analysis been performed appropriately and rigorously?

Reviewer #1: No

Reviewer #2: Yes

Reviewer #3: Yes

3. Have the authors made all data underlying the findings in their manuscript fully available (please refer to the Data Availability Statement at the start of the manuscript PDF file)?

Reviewer #1: No

Reviewer #2: Yes

Reviewer #3: Yes

4. Is the manuscript presented in an intelligible fashion and written in standard English?

Reviewer #1: Yes

Reviewer #2: Yes

Reviewer #3: Yes

5. Review Comments to the Author

Reviewer #1: The authors present an interesting study evaluating differences in the rates of alcohol use and alcohol-related behaviors between IPs and NIPs to guide future EMD interventions.

While I understand and accept their findings, my concern is that their statistical analysis remains incomplete. It would have been great to see some statistical tests or hypothesis testing (chi-square, t-test) evaluating the significance of factors influencing the differences in the rates of alcohol usage and behaviors between the two groups.

I believe their study would be of great benefit in guiding future interventions, but it requires further analysis to be able lend weight to their recommendations. I invite them to resubmit this paper after performing more in depth analysis of the data they have described,

Reviewer #2: The manuscript has explored Alcohol Use among Emergency Medicine Department Patients in Tanzania

Few important points.::

1. The introduction requires substantial improvement. State the rationale of the study carefully and clearly.

2. Please include the conceptual framework of your study.

3. Can you please explain what is AUDIT score and how it is important to include in the study?

4. Can you perform regression analysis?

5. Please use either one or two decimal point?

6. The table are missing captions.

7. There are so many grammatical mistakes and spelling mistakes. please correct them. Also recheck the references?

8. Can you please include policy recommendation.

9. Usman, M., Anand, E., Akhtar, S. N., Umenthala, S. R., Anwar, T., & Unisa, S. (2022). Prevalence and correlates of alcohol and tobacco consumption among research scholars: evidence from a cross-sectional survey of three Indian universities. Drugs, Habits and Social Policy, (ahead-of-print).

Reviewer #3: The authors performed a reanalysis of a previously run study that took place over several months in the emergency room of a single hospital in Tanzania. The authors previously had seen that trauma patients or (injured patients IP) had high rates of alcohol use. They wanted to know about the epidemiology of alcohol use in the non-injured (NIP) patient population. They also compared usage between IP and NIPs.

Introduction:

- None

Methods:

- Patients were excluded if they were unable to consent. It is possible (perhaps even probable) that the rate of alcohol use is different between patients who are able to consent and those who are unable to (even after a waiting period). This exclusion makes sense for the study but should be acknowledged in the limitations.

- It appears that females were oversampled in this study. The reasons given for this are sound but it will affect the reported overall incidence of alcohol use in both IP and NIPs (likely resulting in an under estimate). This should also be acknowledged.

Results/Figures/Tables

- I recommend using a consistent number of decimal points in the tables (probably 1)

- The authors spend a lot of time in their results examining the alcohol use differences between IP and NIPs. I understand that this was the stated purpose of the paper. However, it appears that the proposed intervention is the use of alcohol intervention programs. It is mildly interesting to know the relative use of alcohol between IP and NIPs. What is more of a take home message is that alcohol use is high in both groups and, therefore, interventions can be more broadly applied. The authors do highlight this appropriately in the discussion.

- I do not see any tests of significance between groups in the tables. Was this considered?

Discussion

- None

6. PLOS authors have the option to publish the peer review history of their article (what does this mean?). If published, this will include your full peer review and any attached files.

**Do you want your identity to be public for this peer review?** For information about this choice, including consent withdrawal, please see our Privacy Policy.

Reviewer #1: No

Reviewer #2: **Yes: **Saddaf Naaz Akhtar

Reviewer #3: **Yes: **Geoffrey Anderson

---

## [Editor Report · Decision Letter 1]

25 Sep 2023

Alcohol Use among Emergency Medicine Department Patients in Tanzania: A Comparative Analysis of Injury Versus Non-injury Patients

PGPH-D-23-00562R1

Dear Dr. Staton,

We are pleased to inform you that your manuscript 'Alcohol Use among Emergency Medicine Department Patients in Tanzania: A Comparative Analysis of Injury Versus Non-injury Patients' has been provisionally accepted for publication in PLOS Global Public Health.

Best regards,

Hani Mowafi, M.D., M.P.H.

Academic Editor